# Direct identification of interfacial degradation in blue OLEDs using nanoscale chemical depth profiling

Gustavo F. Trindade [1,4], Soohwan Sul [2,4], Joonghyuk Kim [2,4], Rasmus Havelund[1,4], Anya Eyres[1], Sungjun Park[2], Youngsik Shin[2], Hye Jin Bae[2], Young Mo Sung [2], Lidija Matjacic[1], Yongsik Jung [2], Jungyeon Won[2], Woo Sung Jeon[2], Hyeonho Choi [2], Hyo Sug Lee[2], Jae-Cheol Lee[2,3], Jung-Hwa Kim [2] ✉ & Ian S. Gilmore [1] ✉

Understanding the degradation mechanism of organic light-emitting diodes (OLED) is essential to improve device performance and stability. OLED failure, if not process-related, arises mostly from chemical instability. However, the challenges of sampling from nanoscale organic layers and interfaces with enough analytical information has hampered identification of degradation products and mechanisms. Here, we present a high-resolution diagnostic method of OLED degradation using an Orbitrap mass spectrometer equipped with a gas cluster ion beam to gently desorb nanometre levels of materials, providing unambiguous molecular information with 7-nm depth resolution. We chemically depth profile and analyse blue phosphorescent and thermally-activated delayed fluorescent (TADF) OLED devices at different degradation levels. For OLED devices with short operational lifetimes, dominant chemical degradation mainly relate to oxygen loss of molecules that occur at the interface between emission and electron transport layers (EML/ETL) where exciton distribution is maximised, confirmed by emission zone measurements. We also show approximately one order of magnitude increase in lifetime of devices with slightly modified host materials, which present minimal EML/ETL interfacial degradation and show the method can provide insight for future material and device architecture development.

Organic light-emitting diodes (OLEDs) have become widely used in the display and lighting industry. Their performance, energy efficiency, and manufacturing scalability have enabled commercial success in mobile phones and tablet displays, but the challenge of insufficient device lifetimes still remains for many future applications requiring higher luminance or higher emission energy such as outdoor lighting/displays and blue OLEDs. In particular, blue OLEDs with phosphorescent or thermally-activated delayed fluorescent (TADF) emitters that are highly energy efficient compared to commercially available fluorescence-type blue OLEDs have suffered from decaying luminance which leads to fading and, eventually, failing devices[1]. Significant effort has been dedicated to reducing this degradation to extend the operational device lifetime comparable to phosphorescent red and green OLEDs[2–5].

[1]National Physical Laboratory, NiCE-MSI, Teddington TW11 0LW, UK. [2]Samsung Advanced Institute of Technology, Samsung Electronics Co., Ltd., 130 Samsung-ro, Suwon 16678, Republic of Korea. [3]Present address: Korea Research Institute of Material Property Analysis (KRIMPA), 712, Nongseo-dong 455, Yongin, 17111, Republic of Korea. [4]These authors contributed equally: Gustavo F. Trindade, Soohwan Sul, Joonghyuk Kim, Rasmus Havelund. ✉e-mail: jh1179.kim@samsung.com; ian.gilmore@npl.co.uk

In modern OLED devices, the extrinsic degradation mechanisms mainly related to device processing, such as ingress of water and oxygen, impurities, and poor encapsulation, are well understood and now controlled to the extent that the main mechanisms contributing to the degradation are intrinsic to the device. These include photo- and electro-chemical degradation due to high exciton density, high energy localised excitons, or high energy polarons, electric-field induced inter-layer diffusion of mobile ions, and heat-induced morphological change during device operation[1,6–8]. Much has been learned about the intrinsic mechanisms by varying device structure and material selection. Aziz et al.[9] showed that the transport of holes into a layer of the electroluminescent molecule $Alq_3$ causes the formation of degradation products that quench fluorescence. This explained the improved lifetime for a device with an emissive layer consisting of a hole-transporting material mixed with the $Alq_3$ over a device with separate layers of the same materials. In a recent study of blue OLEDs with mixed host:dopant emissive layers, Kim et al.[10] showed that degradation may occur through reductive quenching of a host exciton to form a radical ion pair (host* + dopant → host•+ + dopant•). Unstable radicals can return to the uncharged ground state through electron transfer from the dopant to the host but this occurs in competition with irreversible degrading reactions, e.g. bond scission, leading to loss of emitters and potentially the formation of fluorescence, or phosphorescence quenching species. Thanks to these research efforts to understand and to reduce degradation, the device lifetimes of red and green OLEDs approaching hundreds of thousand hours are now attainable, although better device efficiency and lifetime are required for blue OLEDs[1]. Nevertheless, it still remains as a scientific challenge to understand the origin and underlying mechanism for device degradation in OLEDs.

Mass spectrometry has been widely applied to identify reaction products or changes in chemical compositions to understand the pathways involved in degradation within organic semiconductor devices. Approaches include analysis from a solution of the small molecules using liquid chromatography-mass spectrometry (LC-MS)[11–13], analysis of processed materials using direct analysis of the device using laser desorption/ionisation time-of flight mass spectrometry (LDI-ToF-MS)[14–16]. Those are useful but give little or no information concerning where in the devices the reactions occur. Recent studies by LDI-ToF-MS and desorption electrospray ionisation mass spectrometry (DESI-MS) demonstrated 3D profiling of aged OLED devices with limited depth resolution of hundreds of nanometres, providing the lateral and axial distribution of degradation products[17,18]. In recent years, secondary ion mass spectrometry (SIMS) has become a powerful capability for chemical depth profiling and 3D imaging of organic electronic devices. The method involves using an energetic focused beam of primary ions directed at the surface. This causes local desorption, or sputtering, of so-called secondary ions from the near-surface (several nanometres) that are detected in a mass spectrometer, typically a time-of-flight (ToF) design. For analysis, a high-energy liquid metal ion source is usually used (e.g., 20 keV $Bi_3^+$) which causes significant molecular fragmentation and creation of radicals[19,20]. A gas cluster ion beam (GCIB) is often used in combination to gently sputter away surface material permitting mass spectra to be obtained from organic electronic devices in a layer-by-layer manner[13,21–24]. However, since only a small fraction of the overall volume of material removed is analysed, the sensitivity is reduced. Whilst ToF-SIMS has made important contributions towards resolving organic device degradation problems[25,26], it has relatively low mass-resolving power ($m/\Delta m$ ~ 10,000), low mass accuracy (10–30 ppm) and a higher signal background, which limits the identification of complex molecules and compounds at low concentration.

We have recently introduced OrbiSIMS[27], which allows sputtering and analysis using the "gentle" gas cluster ion beam. A high-performance Orbitrap™ HF mass spectrometer gives a mass-resolving power, $m/\Delta m$, of ~240,000 (at $m/z$ 200) and a mass accuracy better than 2 ppm[27].

Here, we present a high-resolution diagnostic method of OLED degradation using OrbiSIMS, providing unambiguous molecular information with 7-nm depth resolution. We measured energy-efficient blue phosphorescent and TADF OLED devices and showed that dominant chemical degradation occurred at the interface between emission and electron transport layers (EML/ETL) for devices with short operational lifetimes, whereas interfacial degradation was minimised for devices with longer device lifetimes. Our further analysis using emission zone measurements revealed that exciton distribution within the EML was critical for the device's lifetime. We achieve spectra with quality comparable to high-performance LC-MS from sub-10-nm layer of organic material to enable localisation and identification of reaction products by sampling from nanoscale organic layers and their interfaces in degraded OLED.

## Results

### Chemical depth profiling of blue OLED devices

A series of phosphorescent and TADF blue OLED devices with varying architecture including three host materials, two dopants, and two ETL materials were fabricated as described in Methods. High-resolution mass spectra were acquired from 200 μm × 200 μm areas successively deeper into the devices using a 5 keV argon gas cluster ion beam for sputtering and analysis using an Orbitrap HF mass spectrometer (Fig. 1a). Each spectrum (depth profile data point) consumes a remarkably small amount of material, approximately 0.17 nm thickness of material (equivalent to 6.8 μm³). To measure the best depth resolution under the conditions of this work, a reference device containing a 1-nm-thick blue Ir dopant layer in a host matrix with a total thickness of 50 nm was used (Fig. 1b), We used ISO 20341:2003 based on the method developed by Dowsett et al.[28] to estimate the standard deviation of the Gaussian broadening and decay length depth resolution parameters σ = 2.07 nm and $\lambda_d$ = 2.37 nm respectively with a FWHM depth resolution of 6.2 nm. The low fragmentation from the argon cluster sputtering and high-resolution Orbitrap MS results in high-resolution spectra with quality comparable to LC-MS. The organic materials in each layer give rise to molecular or pseudo-molecular ion peaks that are readily identified based on their mass with sub-ppm accuracy (Supplementary Figs. 1 and 2 and Supplementary Tables 1 and 2). To assess technical repeatability, three repeat depth profiles were recorded from the same device (Supplementary Fig. 3), which demonstrated good measurement repeatability. Depth profiles were obtained from six different device architectures (Fig. 1c) and show that the chemistry of most layers can be resolved for all devices produced in agreement with the schematic of the general device multilayer architecture shown in Fig. 1a. The mCP and TCTA layers cannot be resolved separately as they are only 5 nm thick. The increase of the NPB molecular ion signal ($C_{44}H_{32}N_2^+$) towards the interface with ITO is due to the enhancement of its ionisation probability in the presence of other atoms/molecules (matrix effect)[29,30]. The data for devices with host 1 and dopant 1 presented a lower depth resolution because they were acquired without electronically gating the sputter border during analysis. We have also measured a device using a conventional dual-beam approach and a ToF analyser for comparison. The poorer mass-resolving power and increased fragmentation of the ToF analyser causes some molecular profiles to show high backgrounds in layers where the corresponding materials are not present (Supplementary Fig. 4). The high mass-resolving power of the Orbitrap analyser becomes important when looking for the degradation products described later, as their mass spectrum peaks have low intensities and, in most cases, have intense, neighbouring peaks within a 200-ppm distance from their exact $m/z$ (Supplementary Fig. 5). Whilst the Orbitrap mass analyser in most aspects is superior, the conventional dual-beam approach offers some valuable complementarities such as

rapid 3D-resolved measurements and lower signal variation of fragments in comparison to molecular ions that are often strongly affected by matrix effects[29,30].

## Blue OLEDs chemical degradation

To understand the physical and chemical origin of degradation in the blue OLED devices, we investigated the characteristics and chemical changes in devices of type A (phosphorescent with DBFPO, host 1 phosphorescent emitter), which showed a relatively short lifetime (time to reach 70% of electroluminescence level -0.7 h) as described in Methods. Electroluminescence (EL) as well as device characteristics

were measured, and, for each architecture, three devices were driven at a constant current with a luminance of 1000 cd m$^{-2}$ until their 90% ($T_{90}$),70% ($T_{70}$) and 50% ($T_{50}$) electroluminescence levels were reached. A control device, $T_{100}$, was left pristine. The degradation is summarised in Supplementary Table 3. Each device is driven separately using a sample design shown in Supplementary Fig. 6a, b. Performance characteristics of the $T_{100}$, $T_{90}$, $T_{70}$ and $T_{50}$ devices including external quantum efficiency (EQE), current efficiency, EL spectrum, and current density–voltage plot are provided in Supplementary Fig. 6c–f. The pristine device showed a typical external quantum efficiency of 18.6% at 1000 cd m$^{-2}$ with a maximum EQE of 19.9%,

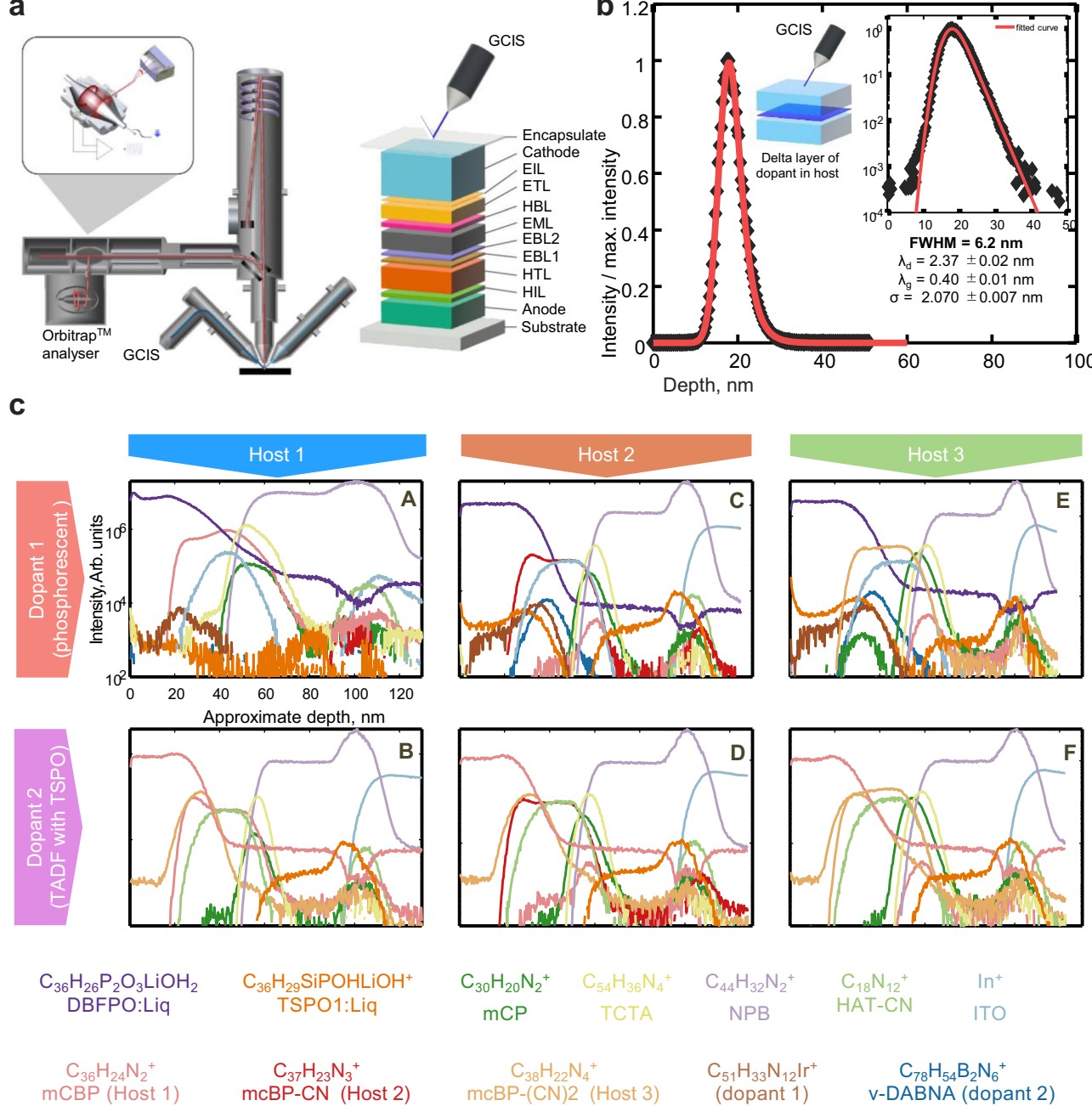

**Fig. 1 | High-resolution chemical depth profiling of blue OLED devices of varying architecture. a** Schematic of the OrbiSIMS instrument showing the gas cluster ion source (GCIS) and the Orbitrap analyser and the layered structure of an OLED device. See Methods for abbreviations of layer names. **b** Depth resolution calculation using a reference 1-nm-thick delta layer of a blue Ir dopant in a host matrix. **c** Depth profiles for all characteristic ions of the various device architectures

arranged in a matrix with different hosts in columns and different dopants in rows (the approximate depth and intensity scales are kept constant across all panels). The capital letters A–F inside each panel represent device architectures with details in Methods. Further details of mass accuracy and mass-resolving power for each ion are given in Supplementary Table 1. Panel (**a**) material adapted from ref. 27.

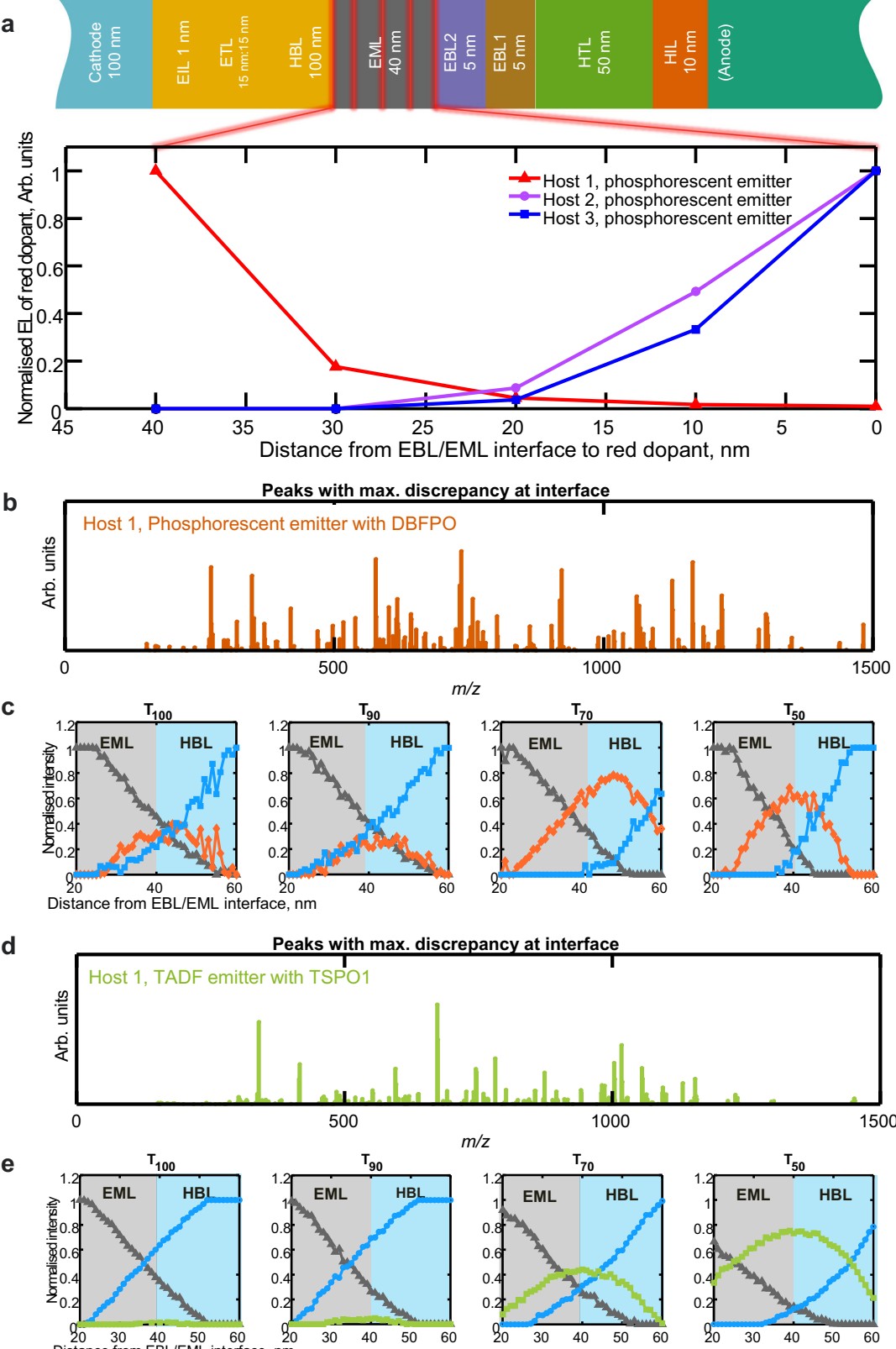

**Fig. 2 | Localisation and identification of blue OLED degradation.**
**a** Electroluminescence (EL) measurements from five test devices for each EML host with monolayer-thick sensing layers of red light-emitting Ir dopants at 0, 10, 20, 30 and 40 nm from the EBL/EML interface. EL intensity is normalised to maximum. Dimensionality reduction via non-negative matrix factorisation of joint datasets $T_{100}$ to $T_{50}$ showing factor intensities across HBL/EML interface and peaks with maximum discrepancy at interface for **b**, **c** phosphorescent devices of type A and **d**, **e** TADF devices of type B. Factor intensities in (**c**) and (**e**) are normalised by the sum of factors and HBL/EML interfaces were determined using non-normalised data. See Methods for abbreviations of layer names.

**Table 1 | Putative assignments for secondary ions related to degradation products in the EML/ETL interface of blue OLED devices**

| m/z | Assignment | Description | Deviation (ppm) |
|---|---|---|---|
| *Phosphorescent (DBFPO based)* | | | |
| 577.1674 | $C_{36}H_{26}P_2O_3LiH_2^+$ | Li.2DBFPO-P(O)Ph2+H | 1.7 |
| 1127.292 | $C_{72}H_{52}P_4O_5Li^+$ | [(DBFPO)$_2$ – O + Li]$^+$ | 0.9 |
| 1219.319 | $C_{78}H_{56}P_4O_6Li^+$ | 2(DBFPO) + Li – Benzene | −0.7 |
| 1019.279 | $C_{66}H_{47}LiO_5P_3^+$ | Liq + DBFPO + Benzene | −0.6 |
| *TADF (TSPO1 based)* | | | |
| 673.247 | $C_{48}H_{38}SiP^+$ | [TSPO1 – O + C$_{12}$H$_9$]$^+$ | −0.9 |
| 340.1329 | $^{13}CC_{23}H_{20}P^+$ | [TSPO1 – SiC$_{12}$H$_9$ – O]$^+$ | −1.3 |
| 674.2507 | $^{13}CC_{47}H_{38}SiP^+$ | [TSPO1 – O + C$_{12}$H$_9$]$^+$ | −0.8 |
| 520.177 | $C_{36}H_{29}SiP^+$ | [TSPO1 – O]$^+$ | −1.4 |
| 416.1641 | $^{13}CC_{29}H_{24}P^+$ | [TSPO1 – SiC$_{12}$H$_9$ – O + C$_6$H$_4$]$^+$ | −1.1 |
| 1055.342 | $C_{72}H_{57}Si_2P_2O^+$ | [2TSPO1 – OH]$^+$ | −0.8 |
| 595.2008 | $SiC_{42}H_{32}P^+$ | [TSPO1 – O + C$_6$H$_3$]$^+$ | −0.9 |

The ions are ordered in terms of their maximum discrepancy at the interface based on PCA loadings results of Supplementary Table 4.

*m/z* mass-to-charge ratio.

indicating that the blue OLED device with an Ir complex emitter utilised most of the injected electrons and holes for phosphorescent emission and its internal quantum efficiency (IQE) was close to unity. Whilst the blue OLED had a reasonable EQE, its operational lifetime was limited to 0.7 h for $T_{70}$.

The EQE of the blue OLED decreased gradually from 19.9% to 13.7% and the driving voltage at a constant luminescence condition increased from 5.9 to 6.2 V upon the device degradation from $T_{100}$ to $T_{70}$ (see Supplementary Table 3 and Supplementary Fig. 6). The EL intensity showed a gradual decrease with device aging, but the normalised EL spectra were identical to each other regardless of the degradation. Although these data provided typical device characteristics and performance evaluation, the underlying origin of the short operational lifetime for the blue device is still elusive, requiring further physical and chemical analysis.

To observe the origin of dominant degradation, we investigated the exciton distribution in the emission zone using the sensing layer method developed by Forrest et al.[31]. For this purpose, we fabricated a series of 5 blue OLED devices (Fig. 2a) with a monolayer-thick sensing layer of red-emitting Ir dopants progressing from the electron blocking layer 2 (EBL2)/EML interface to the hole blocking layer (HBL)/EML interface located at 0, 10, 20, 30 and 40 nm into the EML. The red emission intensity increased towards the HBL/EML layer (Fig. 2a) indicating a large population of excitons close to the HBL. This exciton profile reveals that high energy excitons or polarons from exciton-exciton and exciton-polaron interactions generate most likely close to the HBL/EML interface, where unfavourable chemical reactions with the hole-blocking layer DBFPO cause device degradation[11]. On first inspection, the chemical depth profile from a degraded device has only minor differences from the pristine device ($T_{100}$); however, the spectra from the emissive layers differ in detail that can only be observed due to the high dynamic range and sensitivity of the Orbitrap analyser. Based on the emission layer results (Fig. 2a), we have concatenated the depth profiles of the HBL/EML interface of all four devices ($T_{100}$, $T_{90}$, $T_{70}$ and $T_{50}$) into a single dataset and used the unsupervised data analysis method non-negative matrix factorisation (NMF) to identify groups of characteristic secondary ions at each layer and interface (more details in Methods). Since the spectra only differ in detail and given the high dynamic range and sensitivity of the Orbitrap analyser, only peaks with total intensity across the interface lower than 1000

(0.5% of the max intensity of $2_x10^5$) were considered for the analysis, resulting in a list of 263 peaks. The NMF analysis using three factors identified two consistent factor depth profiles across all devices that represent the HBL and EML layers and a third factor representing the interface but with a trend in intensity from $T_{100}$ to $T_{50}$ (Fig. 2b). All three factors were cross-compared and the secondary ions with highest contributions to the interface factor (in orange) are shown in Fig. 2c. The NMF results allowed for the identification of secondary ions related to interfacial degradation and those are putatively assigned in Table 1. Chemical analysis of OLED degradation has in other studies required further device aging, even to the $T_{10}$ level of degradation[13,21,32]. For comparison with previous studies, the devices were also analysed by traditional LC-MS methods for bulk analysis (Supplementary Fig. 7), but the subtle differences discovered in the OrbiSIMS depth-resolved data were not observed, in particular, the reaction products from aged devices. Therefore, we optimised our method to check degradation products were not created during analysis (more details in Methods and Supplementary Note 2).

The observed degradation-related secondary ions all correspond to potential reaction products of DBFPO (derivatives of DBFPO:Liq), which is in the hole-blocking and electron transport layers (HBL and ETL, respectively). DBFPO consisting of diphenylphosphine oxide and dibenzofuran groups served often as ETL and ambipolar host[18,25]. The molecular formula of the secondary ions indicates that DBFPO oxygen loss is one of the main degradation pathways in the type A device.

The same method was used to identify HBL/EML degradation in devices of type B (TADF with TSPO1, host 1 and a TADF emitter as described in Methods) (0.4 h to reach $T_{70}$ level). NMF results of the concatenated HBL/EML interface data using secondary ions with signal below 1000 intensity also enabled the identification of interfacial degradation products that increased from $T_{100}$ to $T_{50}$ (Fig. 2d, e). The most relevant secondary ions in the interface factors and related to interfacial degradation are putatively assigned in Table 1. Similarly to DBFPO, the interfacial degradation products are related to oxygen loss by the TSPO1 molecule. For the pristine devices ($T_{100}$), the intensities of secondary ions-related interfacial degradation products are much less intense for TSPO1 (Fig. 2e) than for DBFPO (Fig. 2c).

HBL/EML interface depth profiles of characteristic ions for ETL, HBL, EML and degradation products are shown in Fig. 3a, b for $T_{100}$ and $T_{50}$ for both the phosphorescent and TADF type of devices. These clearly show that, for both devices, the only signals that get enhanced from $T_{100}$ to $T_{50}$ are the interfacial degradation-related ones. We observe that the host 1 $T_{100}$ pristine devices have approximately 10% of degradation ion intensity of the $T_{50}$ devices for both emitters, which may be due to beam-induced degradation. This baseline is also found in the NMF analysis in Fig. 2.

## Lifetime and degradation products in blue OLED architectures with different EML hosts

Devices with a change of host in the EML layer were fabricated and tested for degradation, as described in Tables 2 and 3. Lifetime test results showed that a device previously proposed[18] with host 2 and a device with host 3 (mCBP-CN and mCBP-(CN)$_2$ types C and E, respectively in Table 3) presented one order of magnitude longer operational lifetime to reach $T_{50}$ (52.6 h for type C and 80.3 h for type E, with 1.37 h for type A at 1000 cd/m²). The total intensity acquired from nanoscale depth profiling of $C_{72}H_{52}P_4O_5Li^+$ (DBFPO-related degradation product) in Fig. 3c shows that degradation products are present in much less abundance for devices of type C and E in relation to type A, which indicates that a slight change in the host material favourable for electron transport (ET host) may cause a shift of exciton distribution towards opposite to the ETL/EML interface, resulting in less accumulation of degradation products and prolonged device operational lifetime as confirmed by emission zone analyses in the type C and E device in Fig. 2a. A similar trend is observed for TADF devices with the

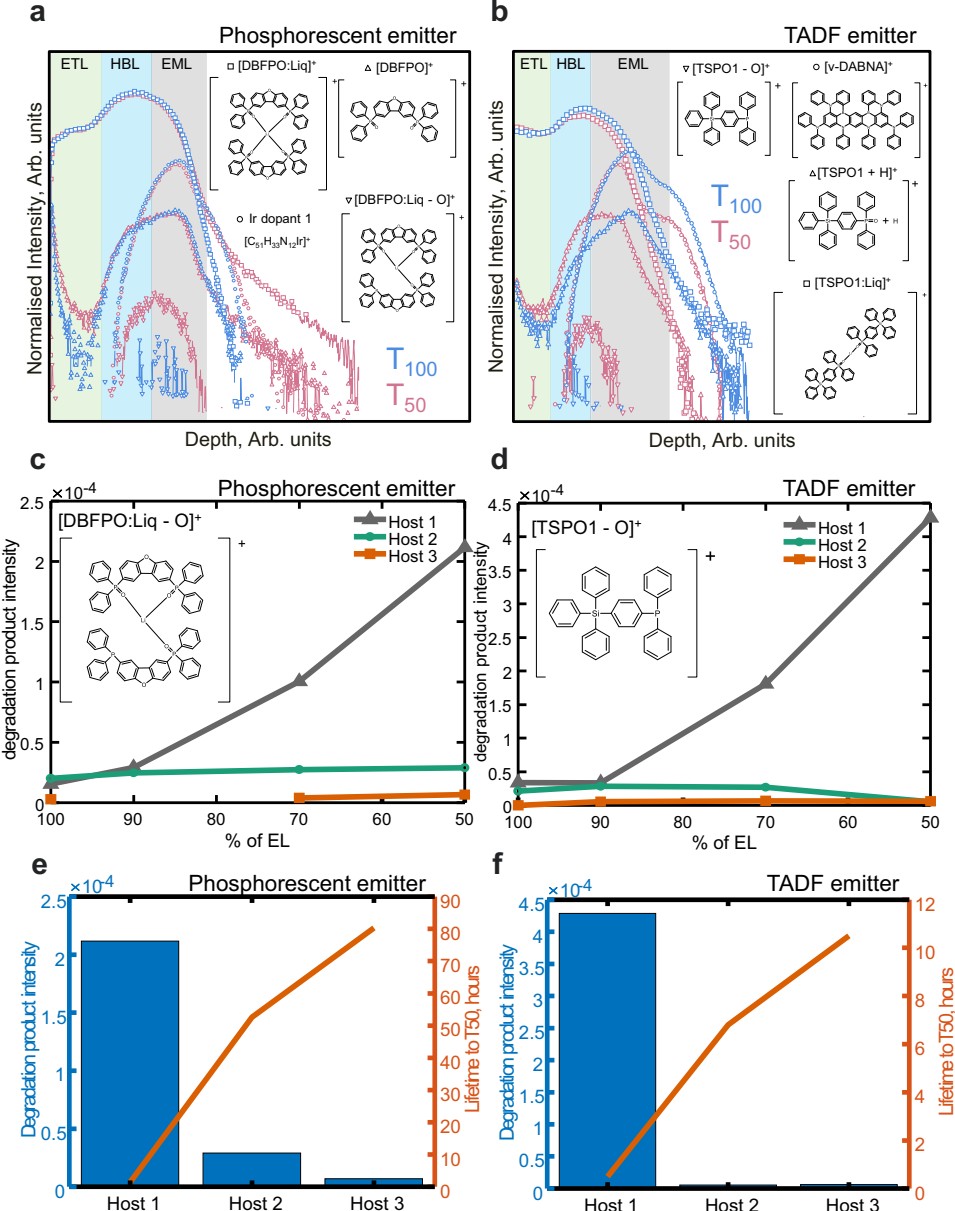

**Fig. 3 | Changes in EML host minimise interfacial degradation and increase device lifetime.** HBL/EML interface chemical depth profiles of characteristic ions for ETL, HBL, EML and degradation products for **a** phosphorescent and **b** TADF device architecture at $T_{100}$ and $T_{50}$ of lifetime. Intensity of secondary ions related to degradation products **c** [DBFPO:Liq]$^+$ and **d** [TSPO1 - O]$^+$ respectively in phosphorescent and TADF devices plotted against % of electroluminescence (EL) for same 3 hosts (devices B, D and F in Table 3 with lifetime to $T_{50}$ of 0.5, 6.8 and 10.5 h, respectively), where the intensity of the secondary ion [M-O]$^+$ (TSPO1-related degradation product) are reduced with the change of hosts with more ET character (Fig. 3d). In fact, Sim et al.[18] showed that the blue OLED device containing mCBP-CN host with the identical device structure except the EML thickness has a long operational lifetime $LT_{50}$ of 431 h at 500 cd/m$^2$ and the excitons within the EML distributed towards the HTL/EML interface. The degradation product intensity for $T_{50}$ compared against lifetime for phosphorescent (Fig. 3e) and TADF (Fig. 3f) devices shows clearly that a change from host 1 to host 3 increases device lifetime These results provide good evidence that our nanoscale chemical depth profiling method can be implemented in the design process and optimisation of new materials to achieve longer lifetimes.

devices with varying EML hosts. Intensity of degradation products **e** [DBFPO:Liq]$^+$ and **f** [TSPO1 - O]$^+$ for $T_{50}$ compared against lifetime to $T_{50}$ respectively for phosphorescent and TADF devices. See Methods for abbreviations of layer names and Table 1 for ion assignments. Molecular structures are for illustration purposes only.

## Discussion

We used nanoscale chemical depth profiling to measure a range of different blue OLED device architectures including TADF and phosphorescent emitters in three types of host materials with different electron-transporting characters. The measurements have a depth resolution of better than 7 nm (less than 1 nm layer of material is consumed per high-resolution mass spectrum), which enabled us to pinpoint the precise information on blue OLED degradation. We investigated the degradation mechanism of highly energy-efficient blue OLEDs in a layer-specific manner and identified molecules characteristic of device degradation, which are located in the ETL and EML and are identified as reaction products of DBFPO or TSPO1 based on their layer-specific location and accurate mass measurement. This has previously not been attainable using any other state-of-the-art

analytical techniques or traditional methods such as LC-MS, ToF-SIMS and so on. The OLED device degradation at the ETL/EML interface was confirmed by a sensing layer method that revealed that the exciton distribution in the emission zone is most intense at the same interface. A clear relationship between the intensity of the identified degradation products, exciton distribution within EML, and device lifetime was found. We showed that the relatively short operational lifetime of a blue phosphorescent OLED with mCBP host originated from the interfacial degradation. We used the method to study devices produced with different host materials with more ET character, which exhibits longer lifetimes and shows that the HBL degradation products were observed but with much less abundance, indicating the utility of our method for guiding material optimisation as well as device architectures to minimise degradation and increase device lifetime.

## Methods

### Materials, device fabrication, and characterisation

Chemicals were purchased from commercial suppliers (Sigma-Aldrich, Wako Pure Chemical Industries, Tokyo Chemical Industry) and they were used without further purification. Most materials were purchased apart from host 2, host 3 and phosphorescent and TADF dopants that were synthesised with details previously described[5,6,10,13,18,33]. Blue phosphorescent and TADF, bottom-emitting OLED devices were fabricated to analyse their performance and degradation mechanism using vacuum evaporating techniques with device structures summarised in Tables 2 and 3.

The organic layers of HIL, HTL, EBL, EML, HBL, ETL and EIL served as hole injection, hole transporting, electron blocking, emitting, hole blocking, electron transporting, and electron injection, respectively. For all devices, the organic and metal layers were deposited consecutively on pre-cleaned ITO glass substrates by using a thermal evaporation system at a pressure less than $1.0 \times 10^{-6}$ mbar. The deposition rates of the organic, Liq, and metal layers were 0.1, 0.01, and $1\,nm\,s^{-1}$, respectively. The active device area was $2 \times 2\,mm^2$ and the devices were encapsulated in a $N_2$-filled glove box prior to all the measurements. The OLED performances were characterised by measuring the current density–voltage–luminance ($J$-$V$-$L$) and electroluminescence spectra using a programmable source meter (Keithley 2400) and a spectrometer (Topcon SR-3AR). The device lifetime measurements (LT90, LT70, LT50) were taken in a constant current mode. LT90, LT70 and LT50 corresponded to the operation time when the percentage luminance decreased to 90%, 70% and 50% at 1000 cd/$m^2$, respectively.

### Secondary ion mass spectrometry

Chemical depth profiles were acquired using two different OrbiSIMS instruments (IONTOF GmbH, Muenster, Germany). These are dual analyser SIMS instruments incorporating an Orbitrap™ mass analyser (Thermo Fisher Scientific, Bremen, Germany) and a time-of-flight (ToF) mass analyser. Secondary ions are extracted through a single set of extraction optics and then an ion-optical switch can send the secondary ions to either of the analysers. The instrument is equipped with a 30 kV Bi Nanoprobe liquid metal ion source and a 5–20 kV gas cluster ion source. This dual-beam dual analyser combination can be operated in multiple modes for spectrometry, 2D imaging and 3D imaging. Depth profiles were obtained in a single beam mode using a 5 keV $Ar_n^+$ gas cluster ion beam (GCIB) as the primary ion beam and varying cluster size $n$ for different measurements. For the analysis of multiple device architectures: $n = 1000$ with 500 V extraction voltage. For the interfacial degradation study, $n = 2500$ with 500 V extraction voltage. For the ion beam-induced degradation study described in Supplementary Note 2, four different beams were used with $n = 697, 1175, 1912$ and 2358 with 500 V extraction voltage. The average primary ion beam current for each depth profile was 100 pA for multiple devices architecture and 50 pA for interface degradation studies. The beam was set to scan an area of 320 µm × 320 µm of which secondary ions were collected from the central 200 µm × 200 µm of the crater. Analysis of secondary ions using the Orbitrap mass analyser with 512 ms injection time, 512 ms transient time for a mass-resolving power of 240,000 at $m/z$ 200. For all OrbiSIMS measurements, the sample target potential was optimised to an average of 130 V by systematically varying it for optimal transmission of molecules of interest[34] and the He collision cell pressure was kept at 0.04 bar in low collisional cooling mode.

**Table 2 | Summary of structures of blue OLED devices where ITO is indium tin oxide, HAT-CN is 1, 4, 5, 8, 9, 11-hexaaza-triphenylene-hexacarbonitrile, NPB is *N*,*N*-di(1-naphthyl)-*N*,*N*'-diphenyl-(1,1'-biphenyl)–4,4'-diamine, TCTA is Tris(4-carbazoyl-9-ylphenyl)amine, mCP is 1,3-Bis(N-carbazolyl) benzene and Liq is lithium quinolinate**

| Layer | Material |
|---|---|
| Anode (150 nm) | ITO |
| HIL (10 nm) | HAT-CN |
| HTL (50 nm) | NPB |
| EBL (5 nm/5 nm) | TCTA/mCP |
| EML (40 nm) | Varied – see Table 3 |
| HBL (10 nm) | Varied – see Table 3 |
| ETL (30 nm) | Varied – see Table 3 |
| EIL (1 nm) | Liq |
| Metal (100 nm) | Al |

**Table 3 | Summary of structures and lifetime measurements of blue OLED devices A to L where the base host material mCBP is 3,3'-Di(9H-carbazol-9-yl)–1,1'-biphenyl, DBFPO is 2,8-bis(diphenylphosphineoxide)-dibenzofuran, TSPO1 is diphenyl[4-(triphenylsilyl)phenyl]phosphine oxide, Liq is lithium quinolinate, v-DABNA is N7,N7,N13,N13,5,9,11,15-octaphenyl-5,9,11,15-tetrahydro-5,9,11,15-tetraaza-19b, 20b-diboradinaphtho[3,2,1-de:1',2',3'-jk] pentacene-7,13-diamine**

| # | Device type | EML | HBL | ETL | LT90 (h) | LT70 (h) | LT50 (h) |
|---|---|---|---|---|---|---|---|
| A | Phosphorescent with DBFPO | mCBP (Host 1):Ir (Dopant 1) | DBFPO | DBFPO:Liq | 0.2 | 0.7 | 1.37 |
| B | TADF with TSPO1 | mCBP (Host 1):v-DABNA (Dopant 2) 5% | TSPO1 | TSPO1:Liq | 0.2 | 0.4 | 0.5 |
| C | Phosphorescent with DBFPO | mCBP-CN (Host 2):Ir (Dopant 1) | DBFPO | DBFPO:Liq | 2.8 | 17.5 | 52.6 |
| D | TADF with TSPO1 | mCBP-CN (Host 2):v-DABNA (Dopant 2) 5% | TSPO1 | TSPO1:Liq | 0.3 | 2 | 6.8 |
| E | Phosphorescent with DBFPO | mCBP-(CN)2 (Host 3):Ir (Dopant 1) | DBFPO | DBFPO:Liq | 5.4 | 29.3 | 80.3 |
| F | TADF with TSPO1 | mCBP-(CN)2 (Host 3):v-DABNA (Dopant 2) 5% | TSPO1 | TSPO1:Liq | 0.8 | 3.8 | 10.5 |

The molecular structure of the Ir dopants is given in Supplementary Table 1.

For comparison, depth profiles were also obtained in a dual-beam mode where the 5 keV $Ar_{2500}^+$ beam was used for sputtering the sample but a 30 keV $Bi_3^+$ (average current 0.15 pA) was used as the analysis beam, and secondary ions were analysed using the ToF analyser (Supplementary Figs. 4 and 5). 21 eV electron flooding was used for charge compensation in all measurements. Prior to depth profiling, the encapsulation was removed from the devices and the Al electrode layer was stripped off using sticky tape. The removal of the Al electrode is required as the GCIB does not effectively sputter metals[35]. To decouple the observed degradation products from any interface degradation induced by the primary ion beam during OrbiSIMS analysis, we have measured devices varying the mean energy per atom in the primary gas cluster ion beam. We observed that ion beams with higher energy per atom, e.g., 5 keV $Ar_{500}^+$ (10 eV/atom), will induce the formation of the same DBFPO degradation products at the HBL/EML interface and have the potential to hinder the study of degradation resulting from the very-low signal-induced degradation from $T_{100}$ to $T_{50}$ levels by creating an extra background signal. The results show that the most appropriate condition is a primary ion beam of 2 eV/atom or below. Further details are given in Supplementary Note 2. This underlines the need for sputtering and analysis with 2 eV/atom or below and hence all measurements to study degradation were done using a 5 keV $Ar_{2500}^+$ ion beam (2 eV/atom).

### Unsupervised machine learning of depth profiles

Unsupervised machine learning for all datasets was carried out using secondary ion masses as the variables and depth levels in-depth profiles as observations. For each dataset, Surface Lab 7.2 (IONTOF GmbH) was used to perform an automated peak search on the total spectra restricted only to peaks with intensity lower than 1000. Peak intensities were then exported for each observation. Non-negative matrix factorisation (NMF) was performed using the simsMVA software[36]. Prior to NMF, data were root-mean scaled to account for non-uniform noise across the mass spectra[37]. Data from multiple depth profiles were reduced to the ETL/EML interface only and arranged in a matrix combining all observations for each type of device for consistent factorisation[38]. The data matrix contained peak intensities in columns and observations in rows. NMF with 3 factors was achieved using a Poisson-based multiplicative update rule algorithm[39]. To identify the *m/z* with maximum discrepancy at the interface, the intensities of the NMF interface factor were subtracted by the mean intensities across factors. The resulting differences were squared and ranked. The full NMF results are available in Supplementary Note 4.

### Photoluminescence spectroscopy

Photoluminescence spectra were taken using a commercial fluorescence spectrometer (Picoquant, Fluotime 300). Pristine and degraded devices with a pixel size of 2 mm × 2 mm were irradiated with a 379-nm diode laser for photoexcitation. The photoluminescence signal from each device was recorded using a PMT detector through a motorised scanning monochromator. At least five independent measurements per each pixel were recorded for statistical analysis.

### Liquid chromatography-mass spectrometry

The liquid chromatographic separations were conducted using a high-performance liquid chromatography system equipped with a C18 column (2.1 × 150 mm) and a photodiode array detector for UV detection (Thermo Fisher Ultimate 300). The HPLC system was coupled to an Orbitrap mass spectrometer (Thermo Fisher Velos Pro) with atmospheric pressure chemical ionisation. For LC-MS sample preparations, a 2 mm × 2 mm OLED pixel was cut and dissolved in 50 µL of tetrahydrofuran, followed by centrifugation to remove electrodes and glass substrate. A 30 µL of the supernatant was transferred for a subsequent LC-MS analysis. The mobile phase flow rate of $H_2O$ and tetrahydrofuran gradient were 0.2 mL min⁻¹ and the injection volume was 3 µL. All mass spectra were collected in positive ion mode.

## Data availability

The data used to create Figs. 1–3 are available in Figshare (https://figshare.com/collections/Direct_identification_of_interfacial_degradation_in_blue_OLEDs_using_nanoscale_chemical_depth_profiling/6904567). The full results of non-negative matrix factorisation presented in Fig. 2 are provided in Supplementary Note 4. The raw OrbiSIMS experimental data of this study are available from the corresponding authors upon request.

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

## Acknowledgements

This work forms part of the Life Science and Health programme of the National Measurement System of the UK Department for Science, Innovation and Technology.

## Author contributions

All authors contributed to the manuscript and approved the final manuscript. The concept was devised by SS, RH, JW, WSJ, J-CL, JK and ISG. RH, JK, GFT, YS and AE acquired the SIMS data. GFT, AE, JK, SS, RH, J-CL and ISG interpreted the data. SS and YMS performed optical device characterisation. LM acquired Orbitrap MS/MS data. SP, YS and SS conducted LC-MS and degradation analysis. JK and HC conducted device fabrication and characterisation. HJB and YJ conducted material synthesis. GFT, SS, RH, AE, LM, JHK and ISG wrote the paper. ISG, JHK and HSL oversaw the study.

## Competing interests

The authors declare no competing interests.
