## [Peer Review File · Nature Communications]

Direct identification of interfacial degradation in blue OLEDs using nanoscale chemical depth profilingREVIEWER COMMENTS

Reviewer #1 (Remarks to the Author):

The issue of OLEDs degradation (or aging) has been addressed many times worldwide. In recent years, ToF-SIMS depth-profiling has been the technique of choice to attempt to solve the long standing question of EL fast decay, thanks to its molecular sensitivity combined with high depth resolution. The OLEDs degradation is often described as a diffusion problem of one or many components. However, very few (if any) papers discuss molecular degradation upon aging leading to the appearance of new species. This paper convincingly identifies degradation ions, which is truly remarkable. This could clearly not be achieved with "regular" dual beam ToF-SIMS but requires the OrbiSIMS. The paper is well written, it is clear and highly significant in both the fields of OLEDs R&D and SIMS analysis.

Here follows a few comments on a few details of the paper that could possibly be improved :

-p6, line 112: it would be useful to justify why the laser irradiation is relevant in this work, compared to the electrical current aging.

-p7, line 134: Ar2500+ cluster impacts are significantly deeper than 0.17 nm (~2-5 nm), so please explain the meaning of this ultrathin 0.17 nm material removal. This obviously is not your depth resolution (7 nm).

-p8. Fig 1b: please explain what are the blue and red dotted lines in the close up profile, as this is missing!

-p8. Fig 1d: the profile is nice, but not "perfect". Many interfaces don't look sharp, signal never reach a steady state and even the layers mCP and TCTA completely overlap as if they were mixed. You might want to be a bit more critical about the quality of the profile.

-It could be interesting to discuss briefly the impact of the Al cathode removal. Are we sure that the cathode plays no role in the aging process, and are we sure the removal does not affect the underlying stack?

-p9, line 169: diphenylphosphine instead of diplenylphosphine

-p12: as all thickness are given in nanometers, avoid Angstroems.

-p17: what is the Ar2500 beam size? Would 3D imaging be possible?

SI:

-Table 1: it may be obvious for some readers, but please define terms in the table (EQEmax, nit, Vd...)

-line 82: define heteroscedasticity

-PCA: the physical meaning of PC1 is missing (PC1 is not really discussed indeed!). Also why do you show PC5, which represents only 6% of the data variance? What about PC2,3,4?

-line 110: Figures 8a and 8b (not 6a and 6b)

Reviewer #2 (Remarks to the Author):

This is an interesting paper and holds significance as a demonstration of depth profiling to detect degradation products in OLED devices. However, the significance is limited, and may not be up to Nature Communications standards. This does not seem to move the field forward significantly relative to previously published demonstrations of chemical degradation detected by depth profiling and mass spectrometry. At times the paper reads more like an advertisement for the authors' "OrbiSIMS" method, but its advantage over other methods like SIMS paired with GCIB sputtering is not clear - perhaps the mass resolution is indeed better but SIMS has given sufficient mass resolution for structure assignments, and no clear comparison is made here - I believe that was the subject of other papers, and may have had significance of its own, but in that case it is no longer a reason for publication in Nature Communications.

The degradation itself is interesting but due to the device and materials choices not likely significant on its own. The device lifetime of T70 = 1.5-2 hours, and not at a particular high luminance, would not lend to industrial or commercial applicability, and calls into question whether any observed degradation pathway would also be operational in a commercially relevant device. Similarly, the use of DBFPO, the material which was found to degrade, is not commonly used in OLED devices despite the authors' claim, which was backed up only by references to two publications studying that molecule's degradation. It is useful in that it allows the authors to detect chemical byproducts of a degradation mechanism, but that mechanism and pathway in general would not necessarily be active in a good device. This shortcoming is acknowledged by the authors, in a way, in the paragraph starting at line 248 in which they study a more stable device but do not find degradation products even of DBFPO at the same levels. The authors claim around line 257 that their techniques help design and optimize new materials, but the better device studied was described in another report, and actually the emission zone study with the red dopant would be more useful in predicting a stable device, moving the emission further into the EML away from any interface.

The chemical degradation product assignments would need to be checked carefully before publication, if this were published. Line 148/Figure 1d mislabels mCBP as DBFPO; Line 199/Figure 2a has m/z=1127 assigned as a Li complex of DBFPO rather than a proper chemical degradation of DBFPO (possibly indicating [Li]Q degradation, certainly the migration mentioned later); Fig 2a also has 577 and 369 assigned as chemical structures that don't make sense (for example 577 would be dicationic as drawn - I think they mean a reduction product of the parent molecule, addition of H₂ to the dibenzofuran

moiety); finally Table 1 doesn't always match Figure 2a, for example $m/z=943$ is labeled LiQ+DBFPO in the table but drawn as Li₂DBFPO-P(O)Ph₂+H in the figure.

The paper is still interesting as another demonstration of the power of depth profiling to give information about chemical degradation mechanisms that could be present in an OLED device, and the authors use excellent controls and supplementary techniques in the photodegradation, photoluminescence, emission zone, etc. The methodology itself is sound, and the DBFPO degradation is reasonable although exact structures need to be cleaned up and perhaps better supported. Generally the work supports the claims, and the detail should suffice for one practiced in the art to replicate the study. But the limitations of the device architecture chosen and the fact that depth profiling with GCIB and mass spectrometry has already been demonstrated in the open literature in a similar way call into question the significance as an addition to this important field.

Reviewer #3 (Remarks to the Author):

In this manuscript the authors present a high-resolution diagnostic method of OLED degradation using an Orbitrap mass spectrometer equipped with a gas cluster ion beam to gently desorb nanometre levels of materials, providing unambiguous molecular information with 7-nm depth resolution. They apply their method in blue phosphorescent OLEDs and show that dominant chemical degradation occurred at the interface between the electron transport layer (i.e., DBFPO) and the emission layers (EML/ETL) where exciton distribution was maximised. By small changes in the EML, they were able to achieve an approximately two orders of magnitude increase in lifetime due to highly suppressed interfacial degradation. The methodology applied here is of interest and the increase in lifetime in blue OLEDs is a very important topic. However, I cannot recommend acceptance of this manuscript in Nature Communications for the following reasons:

1. The topic and work is very specific. Although the technique applied is very sophisticated, it is another highly accurate method to probe interface degradation. I would be more positive if the authors at least examine a series of different devices (with various hole transport materials, electron transport materials and emission layers) in both phosphorescence and TADF devices in order to extract a series of results that would allow the development of general device fabrication rules. As it stands, the manuscript is suitable only for a specialized journal.

2. What are the most important parameters that affect the interface degradation? Is it the differences in energy levels, in mobilities or is it because of some degree of chemical reactions at the interfaces? Again, the manuscript lacks to provide information that could be useful in developing guidelines for the design of more efficient and stable blue OLEDs.

Re: NCOMMS-22-52329: Direct identification of interfacial degradation in blue OLEDs using nanoscale chemical depth profiling

Response to reviewers

Thank you for the reviewers' comments and for the opportunity to submit a major revision. We very much appreciated the reviewers' suggestion to study more devices and to do this we fabricated 6 types of phosphorescent and TADF devices with three different host materials and two different dopants. We have also added an additional set of samples aged to T50, with a total of 24 devices.

Owing to difficulties in obtaining materials and manufacturing pressures resulting from the pandemic these took longer to produce than we anticipated. Unfortunately, the samples did not meet our requirements for reproducibility and so a second set of samples needed to be fabricated adding to the delay.

We are pleased to submit a major revision of the manuscript with substantial additional analysis of different OLED materials. Below we provide a point-by-point response to the reviewers' comments.

REVIEWER COMMENTS

Reviewer #1 (Remarks to the Author):

The issue of OLEDs degradation (or aging) has been addressed many times worldwide. In recent years, ToF-SIMS depth-profiling has been the technique of choice to attempt to solve the long standing question of EL fast decay, thanks to its molecular sensitivity combined with high depth resolution. The OLEDs degradation is often described as a diffusion problem of one or many components. However, very few (if any) papers discuss molecular degradation upon aging leading to the appearance of new species. This paper convincingly identifies degradation ions, which is truly remarkable. This could clearly not be achieved with "regular" dual beam ToF-SIMS but requires the OrbiSIMS. The paper is well written, it is clear and highly significant in both the fields of OLEDs R&D and SIMS analysis.

We are very grateful for the positive comments about the manuscript and we agree that the identification of degradation ions cannot be achieved with dual beam ToF-SIMS methods. We provide some more detail about that in response to questions from the other reviewers.

Here follows a few comments on a few details of the paper that could possibly be improved :

-p6, line 112: it would be useful to justify why the laser irradiation is relevant in this work, compared to the electrical current aging.

Thank you, we agree and now focus the main manuscript entirely on the electrical ageing and have also expanded the study to include a T50 sample set. We have retained the thermal degradation of the device after laser irradiation in Supplementary Note 3.

-p7,line134: Ar2500+ cluster impacts are significantly deeper than 0.17 nm (~2-5 nm), so please explain the meaning of this ultrathin 0.17 nm material removal. This obviously is not your depth resolution (7 nm).

We agree and have now clarified in the text. This relates to the amount of material sampled per data point in the depth profile. It is calculated from the thickness of the device divided by the number of data points in the depth profile. It shows the sensitivity of the method. We now also include the volume of material analysed for comparison with other methods.

-p8. Fig 1b: please explain what are the blue and red dotted lines in the close up profile, as this is missing!

We have now included a response function fit based on an ISO standard to estimate depth resolution, changed the Figure, and added appropriate labels.

-p8. Fig 1d: the profile is nice, but not "perfect". Many interfaces don't look sharp, signal never reach a steady state and even the layers mCP and TCTA completely overlap as if they were mixed. You might want to be a bit more critical about the quality of the profile.

We have now added additional discussion on the depth profile based on quantitative measures of the resolution. In Fig 1b we show a depth resolution of < 7 nm is achieved and this is consistent with not being able to resolve the mCP and TCTA layers separately as they are only 5 nm. We have also included some discussion on matrix artefacts in SIMS where the intensity of one material is enhanced or reduced in the presence of another. This can explain why some materials have less sharp interfaces than others. In addition, following the other reviewer's comments we have fabricated more devices which were analysed using a sputter border mode to electronically gate the depth profile which was not previously available in the NPL prototype OrbiSIMS instrument. This has sharpened the profiles as can be seen.

-It could be interesting to discuss briefly the impact of the Al cathode removal. Are we sure that the cathode plays no role in the aging process, and are we sure the removal does not affect the underlying stack?

In total, including data not used in the manuscript, we have measured more than 40 devices and they all showed comparable depth profiles and no indication of impact of the cathode removal on the underlying stack. The new version of the manuscript may illustrate this better as we have now included measurement of 6 different types of devices. With regards to the ageing process, the cathode is only removed prior to SIMS analysis and any ageing was done in the intact and encapsulated device. We have included a sentence in the manuscript to make this clearer.

-p9, line 169: diphenylphosphine instead of diplenylphosphine

Thank you, now corrected.

-p12: as all thickness are given in nanometers, avoid Angstroems.

Thank you, we have changed to nm.

-p17: what is the Ar2500 beam size? Would 3D imaging be possible?

The 5 keV Ar_{2500}^+ beam spot size used was approximately 20 μm but it can be lowered to approximately 2 μm , with obviously a compromise in beam current. In principle, 3D imaging using the Orbitrap is possible but since we need at least 100 ms per pixel to acquire a spectrum ($\sim 50,000$ mass resolving power for that transient time) the time per 2D image is long and for a 3D image is impractical. We have included some text to highlight this disadvantage and provide more balance to help answer a point from the other reviewer.

We have found that a hybrid approach works well in the OrbiSIMS where we generate the 3D information from high spatial resolution Bi beam imaging with ToF MS and during the GCIB sputter cycle the secondary ions are directed to the Orbitrap giving a complementary high mass resolution depth profile. We have also added a note about this possibility. We do not use it here since we wish to use very low fragmentation conditions that can only be achieved with the GCIB.

SI:

-Table 1: it may be obvious for some readers, but please define terms in the table (EQEmax, nit, Vd...)

Thank you, we have added definitions.

-line 82: define heteroscedasticity

This is now referred to as “non-uniform noise across the mass spectrum”.

-PCA: the physical meaning of PC1 is missing (PC1 is not really discussed indeed!). Also why do you show PC5, which represents only 6% of the data variance? What about PC2,3,4?

-line 110: Figures 8a and 8b (not 6a and 6b)

We have now expanded the supplementary note for completeness and included all PCA results from PC 1 to PC5.

Reviewer #2 (Remarks to the Author):

This is an interesting paper and holds significance as a demonstration of depth profiling to detect degradation products in OLED devices.

We are very grateful to the reviewer for this positive comment on the significance of our work.

However, the significance is limited, and may not be up to Nature Communications standards. This does not seem to move the field forward significantly relative to previously published demonstrations of chemical degradation detected by depth profiling and mass spectrometry. At times the paper reads more like an advertisement for the authors' "OrbiSIMS" method, but its advantage over other methods like SIMS paired with GCIB sputtering is not clear - perhaps the mass resolution is indeed better but SIMS has given sufficient mass resolution for structure assignments, and no clear comparison is made here - I believe that was the subject of other papers, and may have had significance of its own, but in that case it is no longer a reason for publication in Nature Communications.

Many thanks for highlighting this and we are sorry that we did not convey the significance of this study compared with previous studies using dual-beam ToF-SIMS. Aside from the mass resolving power there are two very important differences in the methods.

(1) Fragmentation from the high energy Bi₃ analysis beam in ToF-SIMS.

ToF-SIMS uses a dual beam approach for depth profiling of organics. A low energy per atom gas cluster ion beam (typically ~ 5 eV per atom) is used for sputter removal of the material, which is subsequently analysed using a high energy liquid metal ion beam, usually ~20 keV Bi₃⁺ for analysis (~7000 eV per atom). The reason for this is that the gas cluster ion beam cannot be focussed well in the time domain, which is needed for good mass resolution in the time-of-flight mass spectrum. The liquid metal ion source has a tight time focus allowing mass resolving powers of around 15,000 to be achieved. However, whilst the GCIB is gentle and allows exposure of relatively fresh material the Bi₃⁺ is not and causes strong fragmentation. This problem is well documented in the literature (e.g. Wucher et al <https://doi.org/10.1021/jp8049763> and Muramoto et al <https://doi.org/10.1002/sia.3479>). In the new version of the manuscript, we have studied the effect of the beam cluster size on the identification of degradation products and showed that one must use a very low eV / atom to avoid having the same degradation arising from beam-induced phenomena. This is now described in **Supplementary Note 2**.

Fig R1 Extensive molecular fragmentation caused by Bi_3^+ analysis beam in dual-beam ToF-SIMS depth profiling in contrast to the low fragmentation created by the argon GCIB single-beam OrbiSIMS depth profiling experiment.

(2) The sensitivity of OrbiSIMS is 135 times higher than ToF-SIMS

In dual beam ToF-SIMS most of the material (>99.97%) is removed by the sputtering beam and is not analysed. This is called a low duty cycle and reduces the sensitivity of the method. In a separate study with colleagues at the semiconductor research institute IMEC in Belgium, we have measured this effect using an Sb ion implantation in silicon reference material. The Sb dose is 1×10^{15} atoms/cm². With such a sample we can measure how many atoms of Sb are in the analysed volume and how many Sb ions are detected. The ratio of this is known as the useful yield.

Confidentially (publication in preparation), we show in Fig R2(a) depth profiles of the implant for ToF-SIMS and OrbiSIMS for the SiSb^- ion implant and in Fig R2(b) the cumulative SiSb^- intensity. The OrbiSIMS signal is significantly higher. The calculation of the useful yield (Table R1) shows that the OrbiSIMS is a factor of 135 times higher than for dual beam ToF-SIMS using the same 1 keV Cs^+ sputtering conditions.

ToF-SIMS would not have the sensitivity to detect the OLED degradation products that we find in this study.

Fig R2 Confidential to the reviewers. OrbiSIMS and ToF-SIMS depth profiles of an Sb ion implanted reference sample. Both depth profiles use a 1 keV Cs^+ ion beam (used for inorganic depth profiles). The OrbiSIMS uses the same beam for analysis and the ToF-SIMS uses 15 keV Bi^+ . From Yundong Zhou¹, Alexis Franquet², Valentina

Spampinato², Gustavo F. Trindade¹, Alexander Pirkl³, Wilfried Vandervorst², Paul van der Heide² and Ian Gilmore¹, paper in preparation. ¹NPL, ²IMEC, ³ION-TOF.

Table R1 The useful yield (a direct measure of sensitivity) of the Orbitrap and ToF SIMS measured from the depth profile of an Sb Implant (1×10^{15} atoms / cm²) in silicon using the same sputtering condition of 1 keV Cs⁺. The useful yield for OrbiSIMS is a factor of 135 higher.

	Orbitrap MS	ToF MS
Analysis beam	1k eV Cs ⁺	15 keV Bi ⁺
Sputter beam		1 keV Cs ⁺
Duty cycle	1	2.2×10^{-4}
Analysis area	50 $\mu\text{m} \times 50 \mu\text{m}$	100 $\mu\text{m} \times 100 \mu\text{m}$
Number Sb atoms	2.5×10^{10}	1×10^{11}
Total SiSb ⁻ counts	14021036	414886
Useful yield	5.61×10^{-4}	4.14×10^{-6}

To demonstrate these points, we show in Fig. R3 a comparison of the spectra for 4 degradation products using our OrbiSIMS method and a conventional dual beam ToF-SIMS method. In the OrbiSIMS data the peaks area clearly detected above background signal whilst in the ToF-SIMS data the peaks, if detected at all, are lost in background signals and peak interferences.

Fig. R3 Comparison of spectra using Bi₃⁺/ToF and Ar₂₅₀₀⁺/Orbitrap showing the peaks of 4 blue OLED degradation products.

We hope that the above helps demonstrate the significance of our results compared with published ToF-SIMS studies. We have updated the manuscript to clarify this difference.

We are sorry that the reviewer felt that “at times the paper reads more like an advertisement for the authors' "OrbiSIMS" method”. We have tried to be objective, but our enthusiasm may have got the better of us in places. We have therefore gone through the paper and have also asked an independent colleague at NPL to go through the paper and help us address any issues. For balance, we have also included a comment on the disadvantage of GCIB – Orbitrap MS for 3D imaging, which would require impractical analysis times.

We would like to point out that we have no commercial interest in the OrbiSIMS. It is manufactured by IONTOF.

The degradation itself is interesting but due to the device and materials choices not likely significant on its own. The device lifetime of T70 = 1.5-2 hours, and not at a particular high luminence, would not lend to industrial or commercial applicability, and calls into question whether any observed degradation pathway would also be operational in a commercially relevant device. Similarly, the use of DBFPO, the material which was found to degrade, is not commonly used in OLED devices despite the authors' claim, which was backed up only by references to two publications studying that molecule's degradation.

It is useful in that it allows the authors to detect chemical by products of a degradation mechanism, but that mechanism and pathway in general would not necessarily be active in a good device. This shortcoming is acknowledged by the authors, in a way, in the paragraph starting at line 248 in which they study a more stable device but do not find degradation products even of DBFPO at the same levels. The authors claim around line 257 that their techniques help design and optimize new materials, but the better device studied was described in another report, and actually the emission zone study with the red dopant would be more useful in predicting a stable device, moving the emission further into the EML away from any interface.

We are grateful for the reviewer's comments and also the editors comment that we need to demonstrate our method on a wider range of devices which are of more interest to the community. We have fabricated 6 types phosphorescence and TADF devices with three different host materials and two different dopants to demonstrate the generality of the method. We have now reworded the text to make it clear that the analysis of the more stable device was used to confirm that the amount observed chemical degradation was indeed related to device lifetime. We then provide hypothesis for why one device performs better than the other. This kind of insight on the degradation chemistry goes beyond just identifying a more or less stable device but it can indeed inform future designs. We hope the new version of the manuscript makes this point clearer

The chemical degradation product assignments would need to be checked carefully before publication, if this were published. Line 148/Figure 1d mislabels mCBP as DBFPO; Line 199/Figure 2a has m/z=1127 assigned as a Li complex of DBFPO rather than a proper chemical degradation of DBFPO (possibly indicating [Li]Q degradation, certainly the migration mentioned later); Fig 2a also has 577 and 369 assigned as chemical structures that don't make sense (for example 577 would be dicationic as drawn - I think they mean a reduction product of the parent molecule, addition of H₂ to the dibenzofuran moiety); finally Table 1 doesn't always match Figure 2a, for example m/z=943 is labeled LiQ+DBFPO in the table but drawn as Li.2DBFPO-P(O)Ph₂+H in the figure.

We are very grateful to the reviewer for drawing our attention to these errors. The new version of the manuscript lists some other ions (including TSPO-related degradation) with high accuracy in detection (Table 1) and we only provide structure for two key ions (Figure 3).

The paper is still interesting as another demonstration of the power of depth profiling to give information about chemical degradation mechanisms that could be present in an OLED device, and the authors use excellent controls and supplementary techniques in the photodegradation, photoluminescence, emission zone, etc. The methodology itself is sound, and the DBFPO degradation is reasonable although exact structures need to be cleaned up and perhaps better supported. Generally the work supports the claims, and the detail should suffice for one practiced in the art to replicate the study.

We thank the reviewer for these supportive comments.

But the limitations of the device architecture chosen and the fact that depth profiling with GCIB and mass spectrometry has already been demonstrated in the open literature in a similar way call into question the significance as an addition to this important field.

We understand the reviewer's concern and we hope that the fabrication and analysis of the new devices and the improved explanation in this letter and in the manuscript of the benefits of the GCIB – Orbitrap MS method compared with previously published dual-beam ToF-SIMS depth profiles helps demonstrate the significance of this study beyond published work. This is the first paper to show LC-MS quality high-resolution mass spectrometry (Orbitrap) of molecular layers with low fragmentation (GCIB with < 2 eV / atom) with a depth-resolution of < 7 nm.

Reviewer #3 (Remarks to the Author):

In this manuscript the authors present a high-resolution diagnostic method of OLED degradation using an Orbitrap mass spectrometer equipped with a gas cluster ion beam to gently desorb nanometre levels of materials, providing unambiguous molecular information with 7-nm depth resolution. They apply their method in blue phosphorescent OLEDs and show that dominant chemical degradation occurred at the interface between the electron transport layer (i.e., DBFPO) and the emission layers (EML/ETL) where exciton distribution was maximised. By small changes in the EML, they were able to achieve an approximately two orders of magnitude increase in lifetime due to highly suppressed interfacial degradation. The methodology applied here is of interest and the increase in lifetime in blue OLEDs is a very important topic

We are grateful to the reviewer for this nice synopsis and positive comments about our work and the importance of increased lifetime for blue OLEDs.

However, I cannot recommend acceptance of this manuscript in Nature Communications for the following reasons:

We hope that the major revisions we have made including fabrication of new and more relevant devices with additional analysis and interpretation along with an improved explanation of the novelty of our method in comparison with previously published dual-beam ToF-SIMS studies will help convince the reviewer that the manuscript is suitable for Nature Communications. As we demonstrate in Figs R1, R2, R3, ToF-SIMS does not have the sensitivity or specificity to detect or identify the interfacial degradation products. We provide additional comments below.

1. The topic and work is very specific. Although the technique applied is very sophisticated, it is another highly accurate method to probe interface degradation. I would be more positive if the authors at least examine a series of different devices (with various hole transport materials, electron transport materials and emissive layers) in both phosphorescence and TADF devices in order to extract a series of results that would allow the development of general device fabrication rules.

We thank the reviewer for their suggestion and have fabricated 6 phosphorescence and TADF devices with three different host materials and two different electron transporting materials to demonstrate the generality of the method. In particular, three host materials have different characters of hole and electron mobility, having more ET character with increasing the number of cyano groups, which changes the exciton distribution within the emissive layer. We demonstrate the utility of our method more generally and we hope the reviewer is now more positive about the suitability of the manuscript for Nature Communications.

2. What are the most important parameters that affect the interface degradation? Is it the differences in energy levels, in mobilities or is it because of some degree of chemical reactions at the interfaces? Again, the manuscript lacks to provide information that could be useful in developing guidelines for the design of more efficient and stable blue OLEDs.

We thank the reviewer. Three degradation mechanisms in current state-of-the-art OLED devices arises from the formation of exciton localization, exciton-exciton, and exciton-polaron (*J. Appl. Phys.*, 2008, 103, 044509, Forrest Group). Among these degradation mechanisms, exciton-exciton (E-E) and exciton-polaron (E-P) interactions are most important, because they are dominant in high fluence OLED applications. Organic molecules with 'hot' or high energy excitations or polarons originating from E-E or E-P interactions possesses electrons in higher excited states that are often excited dissociative electronic states leading to chemical bond cleavage. It is the emissive layer (EML) that exciton or polarons (negative or positive ions due to holes or electrons) are most abundant and, therefore, the control of exciton distribution within the EML is most important. So, the most important parameter that affect the interface degradation is the exciton distribution within the EML or charge balance within the layer. When the exciton populates close to the ETL/EML interface, it is most probable that high energy negative polarons generates, which leads dissociative channels of chemical bonds or other chemical reactions.

Yours sincerely,

Professor Ian S. Gilmore FMedSci

NPL Senior Fellow

Director of National Centre of Excellence in Mass Spectrometry Imaging

Email: ian.gilmore@npl.co.uk

Tel: +44 (0)208 943 6922

REVIEWERS' COMMENTS

Reviewer #1 (Remarks to the Author):

My comments on the first version of the paper were already quite positive. It is true that the work presented here is rather specific, using the OrbiSIMS which is still known mostly by SIMS specialists, on specific types of OLEDs. However, you have significantly broadened the scope of the paper by fabricating 5 more devices, as was requested by the other reviewers.

I appreciate that you have carefully addressed all my comments, as well as the more critical ones from the other reviewers.

I have no reason to change my mind on your paper. It has in fact, in my opinion, been significantly improved thanks to considerable further measurements.

I still have a question related to fig.2c, where the "orange line" (degradation product contribution) is already quite high on the pristine device. How do you explain that? Is there already chemical degradation on non driven samples? You could add a short discussion on that if the paper gets published.

Reviewer #2 (Remarks to the Author):

The authors have addressed my concerns, between the response they submitted and the changes to the manuscript. I would still encourage the editors to consider whether this manuscript in itself holds the impact expected for a Nature Communications publication. I do see now that the new technique represents a significant advance in the ability to detect degradation products in OLED devices, but some of the demonstration of that is in confidential remarks to be part of another manuscript. The main advance to the field in this manuscript is a now much better established and clearly discussed degradation mechanism, but it remains a mechanism not likely to be present in a well-designed device, including some of the better-performing devices in this manuscript.

I appreciate the work the authors put in to this manuscript and support its publication, and I leave it to the editors to determine whether this is the right place.

Reviewer #3 (Remarks to the Author):

The authors have successfully addressed the comments previously raised by the reviewers. They have demonstrated the interfacial degradation of OLEDs using nanoscale depth profiling which also leads to enhance lifetime in blue OLEDs. I find the current version suitable for Nature Communications and I therefore suggest acceptance.

2nd REVISION

Response to reviewers

Thank you for the reviewers' positive comments. We are pleased to submit a revision of the manuscript and below we provide a point-by-point response to the reviewers' comments.

REVIEWERS COMMENTS

Reviewer #1 (Remarks to the Author):

My comments on the first version of the paper were already quite positive. It is true that the work presented here is rather specific, using the OrbiSIMS which is still known mostly by SIMS specialists, on specific types of OLEDs. However, you have significantly broadened the scope of the paper by fabricating 5 more devices, as was requested by the other reviewers. I appreciate that you have carefully addressed all my comments, as well as the more critical ones from the other reviewers. I have no reason to change my mind on your paper. It has in fact, in my opinion, been significantly improved thanks to considerable further measurements.

We are very grateful for the reviewer's encouraging comments and hope this work will contribute to the understanding of blue OLED's degradation as well as to the increasingly uptake of OrbiSIMS to study organic electronic devices.

I still have a question related to fig.2c, where the "orange line" (degradation product contribution) is already quite high on the pristine device. How do you explain that? Is there already chemical degradation on non driven samples? You could add a short discussion on that if the paper gets published.

Thank you for this helpful comment. We use the multivariate analysis to identify trends in the data and associated mass peaks, but it is better to look at the direct intensities for quantitative analysis. In Fig 3c and d we see that there is a relatively small (~10%) intensity for the degradation ion for host 1 at T100 compared with T50 for both emitters. This could be due to beam-induced degradation. We have included the following discussion in line 268: "We observe that the host 1 T100 pristine devices have approximately 10% of degradation ion intensity of the T50 devices for both emitters, which may be due to beam-induced degradation. This baseline is also found in the NMF analysis in Figure 2."

Reviewer #2 (Remarks to the Author):

The authors have addressed my concerns, between the response they submitted and the changes to the manuscript. I would still encourage the editors to consider whether this manuscript in itself holds the impact expected for a Nature Communications publication. I do see now that the new technique represents a significant advance in the ability to detect degradation products in OLED devices, but some of the demonstration of that is in confidential remarks to be part of another manuscript. The main advance to the field in this manuscript is a now much better established and clearly discussed degradation mechanism, but it remains a mechanism not likely to be present in a well-designed device, including some of the better-performing devices in this manuscript. I appreciate the work the authors put in to this manuscript and support its publication, and I leave it to the editors to determine whether this is the right place.

We are very grateful for the reviewer's comment and pleased that we have made the main advance to the field clearer.

Reviewer #3 (Remarks to the Author):

The authors have successfully addressed the comments previously raised by the reviewers. They have demonstrated the interfacial degradation of OLEDs using nanoscale depth profiling which also leads to enhance lifetime in blue OLEDs. I find the current version suitable for Nature Communications and I therefore suggest acceptance.

We are very grateful for the reviewer's positive comments.

1ST REVISION

Response to reviewers

Thank you for the reviewers' comments and for the opportunity to submit a major revision. We very much appreciated the reviewers' suggestion to study more devices and to do this we fabricated 6 types of phosphorescent and TADF devices with three different host materials and two different dopants. We have also added an additional set of samples aged to T50, with a total of 24 devices.

Owing to difficulties in obtaining materials and manufacturing pressures resulting from the pandemic these took longer to produce than we anticipated. Unfortunately, the samples did not meet our requirements for reproducibility and so a second set of samples needed to be fabricated adding to the delay.

We are pleased to submit a major revision of the manuscript with substantial additional analysis of different OLED materials. Below we provide a point-by-point response to the reviewers' comments.

REVIEWERS COMMENTS

Reviewer #1 (Remarks to the Author):

The issue of OLEDs degradation (or aging) has been addressed many times worldwide. In recent years, ToF-SIMS depth-profiling has been the technique of choice to attempt to solve the long standing question of EL fast decay, thanks to its molecular sensitivity combined with high depth resolution. The OLEDs degradation is often described as a diffusion problem of one or many components. However, very few (if any) papers discuss molecular degradation upon aging leading to the appearance of new species. This paper convincingly identifies degradation ions, which is truly remarkable. This could clearly not be achieved with "regular" dual beam ToF-SIMS but requires the OrbiSIMS. The paper is well written, it is clear and highly significant in both the fields of OLEDs R&D and SIMS analysis.

We are very grateful for the positive comments about the manuscript and we agree that the identification of degradation ions cannot be achieved with dual beam ToF-SIMS methods. We provide some more detail about that in response to questions from the other reviewers.

Here follows a few comments on a few details of the paper that could possibly be improved :

-p6, line 112: it would be useful to justify why the laser irradiation is relevant in this work, compared to the electrical current aging.

Thank you, we agree and now focus the main manuscript entirely on the electrical ageing and have also expanded the study to include a T50 sample set. We have retained the thermal degradation of the device after laser irradiation in Supplementary Note 3.

-p7,line134: Ar2500+ cluster impacts are significantly deeper than 0.17 nm (~2-5 nm), so please explain the meaning of this ultrathin 0.17 nm material removal. This obviously is not your depth resolution (7 nm).

We agree and have now clarified in the text. This relates to the amount of material sampled per data point in the depth profile. It is calculated from the thickness of the device divided by the number of data points in the depth profile. It shows the sensitivity of the method. We now also include the volume of material analysed for comparison with other methods.

-p8. Fig 1b: please explain what are the blue and red dotted lines in the close up profile, as this is missing!

We have now included a response function fit based on an ISO standard to estimate depth resolution, changed the Figure, and added appropriate labels.

-p8. Fig 1d: the profile is nice, but not "perfect". Many interfaces don't look sharp, signal never reach a steady state and even the layers mCP and TCTA completely overlap as if they were mixed. You might want to be a bit more critical about the quality of the profile.

We have now added additional discussion on the depth profile based on quantitative measures of the resolution. In Fig 1b we show a depth resolution of < 7 nm is achieved and this is consistent with not being able to resolve the mCP and TCTA layers separately as they are only 5 nm. We have also included some discussion on matrix artefacts in SIMS where the intensity of one material is enhanced or reduced in the presence of another. This can explain why some materials have less sharp interfaces than others. In addition, following the other reviewer's comments we have fabricated more devices which were analysed using a sputter border mode to electronically gate the depth profile which was not previously available in the NPL prototype OrbiSIMS instrument. This has sharpened the profiles as can be seen.

-It could be interesting to discuss briefly the impact of the Al cathode removal. Are we sure that the cathode plays no role in the ageing process, and are we sure the removal does not affect the underlying stack?

In total, including data not used in the manuscript, we have measured more than 40 devices and they all showed comparable depth profiles and no indication of impact of the cathode removal on the underlying stack. The new version of the manuscript may illustrate this better as we have now included measurement of 6 different types of devices. With regards to the ageing process, the cathode is only removed prior to SIMS analysis and any ageing was done in the intact and encapsulated device. We have included a sentence in the manuscript to make this clearer.

-p9, line 169: diphenylphosphine instead of diplenylphosphine

Thank you, now corrected.

-p12: as all thickness are given in nanometers, avoid Angstroems.

Thank you, we have changed to nm.

-p17: what is the Ar2500 beam size? Would 3D imaging be possible?

The 5 keV Ar₂₅₀₀⁺ beam spot size used was approximately 20 μm but it can be lowered to approximately 2 μm, with obviously a compromise in beam current. In principle, 3D imaging using the Orbitrap is possible but since we need at least 100 ms per pixel to acquire a spectrum (~ 50,000 mass resolving power for that transient time) the time per 2D image is long and for a 3D image is impractical. We have included some text to highlight this disadvantage and provide more balance to help answer a point from the other reviewer.

We have found that a hybrid approach works well in the OrbiSIMS where we generate the 3D information from high spatial resolution Bi beam imaging with ToF MS and during the GCIB sputter cycle the secondary ions are

directed to the Orbitrap giving a complementary high mass resolution depth profile. We have also added a note about this possibility. We do not use it here since we wish to use very low fragmentation conditions that can only be achieved with the GCIB.

SI:

-Table 1: it may be obvious for some readers, but please define terms in the table (EQEmax, nit, Vd...)

Thank you, we have added definitions.

-line 82: define heteroscedasticity

This is now referred to as “non-uniform noise across the mass spectrum”.

-PCA: the physical meaning of PC1 is missing (PC1 is not really discussed indeed!). Also why do you show PC5, which represents only 6% of the data variance? What about PC2,3,4?

-line 110: Figures 8a and 8b (not 6a and 6b)

We have now expanded the supplementary note for completeness and included all PCA results from PC 1 to PC5.

Reviewer #2 (Remarks to the Author):

This is an interesting paper and holds significance as a demonstration of depth profiling to detect degradation products in OLED devices.

We are very grateful to the reviewer for this positive comment on the significance of our work.

However, the significance is limited, and may not be up to Nature Communications standards. This does not seem to move the field forward significantly relative to previously published demonstrations of chemical degradation detected by depth profiling and mass spectrometry. At times the paper reads more like an advertisement for the authors' "OrbiSIMS" method, but its advantage over other methods like SIMS paired with GCIB sputtering is not clear - perhaps the mass resolution is indeed better but SIMS has given sufficient mass resolution for structure assignments, and no clear comparison is made here - I believe that was the subject of other papers, and may have had significance of its own, but in that case it is no longer a reason for publication in Nature Communications.

Many thanks for highlighting this and we are sorry that we did not convey the significance of this study compared with previous studies using dual-beam ToF-SIMS. Aside from the mass resolving power there are two very important differences in the methods.

(1) Fragmentation from the high energy Bi₃ analysis beam in ToF-SIMS.

ToF-SIMS uses a dual beam approach for depth profiling of organics. A low energy per atom gas cluster ion beam (typically ~ 5 eV per atom) is used for sputter removal of the material, which is subsequently analysed using a high energy liquid metal ion beam, usually ~20 keV Bi₃⁺ for analysis (~7000 eV per atom). The reason for this is that the gas cluster ion beam cannot be focussed well in the time domain, which is needed for good mass resolution in the time-of-flight mass spectrum. The liquid metal ion source has a tight time focus allowing mass resolving powers of around 15,000 to be achieved. However, whilst the GCIB is gentle and allows exposure of relatively fresh material the Bi₃⁺ is not and causes strong fragmentation. This problem is well documented in the literature (e.g. Wucher et al <https://doi.org/10.1021/jp8049763> and Muramoto et al <https://doi.org/10.1002/sia.3479>). In the new version of the manuscript, we have studied the effect of the beam cluster size on the identification of degradation products and showed that one must use a very low eV / atom to avoid having the same degradation arising from beam-induced phenomena. This is now described in **Supplementary Note 2**.

Fig R1 Extensive molecular fragmentation caused by Bi_3^+ analysis beam in dual-beam ToF-SIMS depth profiling in contrast to the low fragmentation created by the argon GCIB single-beam OrbiSIMS depth profiling experiment.

(2) The sensitivity of OrbiSIMS is 135 times higher than ToF-SIMS

In dual beam ToF-SIMS most of the material (>99.97%) is removed by the sputtering beam and is not analysed. This is called a low duty cycle and reduces the sensitivity of the method. In a separate study with colleagues at the semiconductor research institute IMEC in Belgium, we have measured this effect using an Sb ion implantation in silicon reference material. The Sb dose is 1×10^{15} atoms/cm². With such a sample we can measure how many atoms of Sb are in the analysed volume and how many Sb ions are detected. The ratio of this is known as the useful yield.

Confidentially (publication in preparation), we show in Fig R2(a) depth profiles of the implant for ToF-SIMS and OrbiSIMS for the SiSb^- ion implant and in Fig R2(b) the cumulative SiSb^- intensity. The OrbiSIMS signal is significantly higher. The calculation of the useful yield (Table R1) shows that the OrbiSIMS is a factor of 135 times higher than for dual beam ToF-SIMS using the same 1 keV Cs^+ sputtering conditions.

ToF-SIMS would not have the sensitivity to detect the OLED degradation products that we find in this study.

Fig R2 Confidential to the reviewers. OrbiSIMS and ToF-SIMS depth profiles of an Sb ion implanted reference sample. Both depth profiles use a 1 keV Cs^+ ion beam (used for inorganic depth profiles). The OrbiSIMS uses the same beam for analysis and the ToF-SIMS uses 15 keV Bi^+ . From Yundong Zhou¹, Alexis Franquet², Valentina

Spampinato², Gustavo F. Trindade¹, Alexander Pirkl³, Wilfried Vandervorst², Paul van der Heide² and Ian Gilmore¹, paper in preparation. ¹NPL, ²IMEC, ³ION-TOF.

Table R1 The useful yield (a direct measure of sensitivity) of the Orbitrap and ToF SIMS measured from the depth profile of an Sb Implant (1×10^{15} atoms / cm²) in silicon using the same sputtering condition of 1 keV Cs⁺. The useful yield for OrbiSIMS is a factor of 135 higher.

	Orbitrap MS	ToF MS
Analysis beam	1k eV Cs ⁺	15 keV Bi ⁺
Sputter beam		1 keV Cs ⁺
Duty cycle	1	2.2×10^{-4}
Analysis area	50 $\mu\text{m} \times 50 \mu\text{m}$	100 $\mu\text{m} \times 100 \mu\text{m}$
Number Sb atoms	2.5×10^{10}	1×10^{11}
Total SiSb ⁻ counts	14021036	414886
Useful yield	5.61×10^{-4}	4.14×10^{-6}

To demonstrate these points, we show in Fig. R3 a comparison of the spectra for 4 degradation products using our OrbiSIMS method and a conventional dual beam ToF-SIMS method. In the OrbiSIMS data the peaks area clearly detected above background signal whilst in the ToF-SIMS data the peaks, if detected at all, are lost in background signals and peak interferences.

Fig. R3 Comparison of spectra using Bi₃⁺/ToF and Ar₂₅₀₀⁺/Orbitrap showing the peaks of 4 blue OLED degradation products.

We hope that the above helps demonstrate the significance of our results compared with published ToF-SIMS studies. We have updated the manuscript to clarify this difference.

We are sorry that the reviewer felt that “at times the paper reads more like an advertisement for the authors' "OrbiSIMS" method”. We have tried to be objective, but our enthusiasm may have got the better of us in places. We have therefore gone through the paper and have also asked an independent colleague at NPL to go through the paper and help us address any issues. For balance, we have also included a comment on the disadvantage of GCIB – Orbitrap MS for 3D imaging, which would require impractical analysis times.

We would like to point out that we have no commercial interest in the OrbiSIMS. It is manufactured by IONTOF.

The degradation itself is interesting but due to the device and materials choices not likely significant on its own. The device lifetime of T70 = 1.5-2 hours, and not at a particular high luminence, would not lend to industrial or commercial applicability, and calls into question whether any observed degradation pathway would also be operational in a commercially relevant device. Similarly, the use of DBFPO, the material which was found to degrade, is not commonly used in OLED devices despite the authors' claim, which was backed up only by references to two publications studying that molecule's degradation.

It is useful in that it allows the authors to detect chemical by products of a degradation mechanism, but that mechanism and pathway in general would not necessarily be active in a good device. This shortcoming is acknowledged by the authors, in a way, in the paragraph starting at line 248 in which they study a more stable device but do not find degradation products even of DBFPO at the same levels. The authors claim around line 257 that their techniques help design and optimize new materials, but the better device studied was described in another report, and actually the emission zone study with the red dopant would be more useful in predicting a stable device, moving the emission further into the EML away from any interface.

We are grateful for the reviewer's comments and also the editors comment that we need to demonstrate our method on a wider range of devices which are of more interest to the community. We have fabricated 6 types phosphorescence and TADF devices with three different host materials and two different dopants to demonstrate the generality of the method. We have now reworded the text to make it clear that the analysis of the more stable device was used to confirm that the amount observed chemical degradation was indeed related to device lifetime. We then provide hypothesis for why one device performs better than the other. This kind of insight on the degradation chemistry goes beyond just identifying a more or less stable device but it can indeed inform future designs. We hope the new version of the manuscript makes this point clearer

The chemical degradation product assignments would need to be checked carefully before publication, if this were published. Line 148/Figure 1d mislabels mCBP as DBFPO; Line 199/Figure 2a has m/z=1127 assigned as a Li complex of DBFPO rather than a proper chemical degradation of DBFPO (possibly indicating [Li]Q degradation, certainly the migration mentioned later); Fig 2a also has 577 and 369 assigned as chemical structures that don't make sense (for example 577 would be dicationic as drawn - I think they mean a reduction product of the parent molecule, addition of H₂ to the dibenzofuran moiety); finally Table 1 doesn't always match Figure 2a, for example m/z=943 is labeled LiQ+DBFPO in the table but drawn as Li.2DBFPO-P(O)Ph₂+H in the figure.

We are very grateful to the reviewer for drawing our attention to these errors. The new version of the manuscript lists some other ions (including TSPO-related degradation) with high accuracy in detection (Table 1) and we only provide structure for two key ions (Figure 3).

The paper is still interesting as another demonstration of the power of depth profiling to give information about chemical degradation mechanisms that could be present in an OLED device, and the authors use excellent controls and supplementary techniques in the photodegradation, photoluminescence, emission zone, etc. The methodology itself is sound, and the DBFPO degradation is reasonable although exact structures need to be cleaned up and perhaps better supported. Generally the work supports the claims, and the detail should suffice for one practiced in the art to replicate the study.

We thank the reviewer for these supportive comments.

But the limitations of the device architecture chosen and the fact that depth profiling with GCIB and mass spectrometry has already been demonstrated in the open literature in a similar way call into question the significance as an addition to this important field.

We understand the reviewer's concern and we hope that the fabrication and analysis of the new devices and the improved explanation in this letter and in the manuscript of the benefits of the GCIB – Orbitrap MS method compared with previously published dual-beam ToF-SIMS depth profiles helps demonstrate the significance of this study beyond published work. This is the first paper to show LC-MS quality high-resolution mass spectrometry (Orbitrap) of molecular layers with low fragmentation (GCIB with < 2 eV / atom) with a depth-resolution of < 7 nm.

Reviewer #3 (Remarks to the Author):

In this manuscript the authors present a high-resolution diagnostic method of OLED degradation using an Orbitrap mass spectrometer equipped with a gas cluster ion beam to gently desorb nanometre levels of materials, providing unambiguous molecular information with 7-nm depth resolution. They apply their method in blue phosphorescent OLEDs and show that dominant chemical degradation occurred at the interface between the electron transport layer (i.e., DBFPO) and the emission layers (EML/ETL) where exciton distribution was maximised. By small changes in the EML, they were able to achieve an approximately two orders of magnitude increase in lifetime due to highly suppressed interfacial degradation. The methodology applied here is of interest and the increase in lifetime in blue OLEDs is a very important topic

We are grateful to the reviewer for this nice synopsis and positive comments about our work and the importance of increased lifetime for blue OLEDs.

However, I cannot recommend acceptance of this manuscript in Nature Communications for the following reasons:

We hope that the major revisions we have made including fabrication of new and more relevant devices with additional analysis and interpretation along with an improved explanation of the novelty of our method in comparison with previously published dual-beam ToF-SIMS studies will help convince the reviewer that the manuscript is suitable for Nature Communications. As we demonstrate in Figs R1, R2, R3, ToF-SIMS does not have the sensitivity or specificity to detect or identify the interfacial degradation products. We provide additional comments below.

1. The topic and work is very specific. Although the technique applied is very sophisticated, it is another highly accurate method to probe interface degradation. I would be more positive if the authors at least examine a series of different devices (with various hole transport materials, electron transport materials and emissive layers) in both phosphorescence and TADF devices in order to extract a series of results that would allow the development of general device fabrication rules.

We thank the reviewer for their suggestion and have fabricated 6 phosphorescence and TADF devices with three different host materials and two different electron transporting materials to demonstrate the generality of the method. In particular, three host materials have different characters of hole and electron mobility, having more ET character with increasing the number of cyano groups, which changes the exciton distribution within the emissive layer. We demonstrate the utility of our method more generally and we hope the reviewer is now more positive about the suitability of the manuscript for Nature Communications.

2. What are the most important parameters that affect the interface degradation? Is it the differences in energy levels, in mobilities or is it because of some degree of chemical reactions at the interfaces? Again, the manuscript lacks to provide information that could be useful in developing guidelines for the design of more efficient and stable blue OLEDs.

We thank the reviewer. Three degradation mechanisms in current state-of-the-art OLED devices arises from the formation of exciton localization, exciton-exciton, and exciton-polaron (*J. Appl. Phys.*, 2008, 103, 044509, Forrest Group). Among these degradation mechanisms, exciton-exciton (E-E) and exciton-polaron (E-P) interactions are most important, because they are dominant in high fluence OLED applications. Organic molecules with 'hot' or high energy excitations or polarons originating from E-E or E-P interactions possesses electrons in higher excited states that are often excited dissociative electronic states leading to chemical bond cleavage. It is the emissive layer (EML) that exciton or polarons (negative or positive ions due to holes or electrons) are most abundant and, therefore, the control of exciton distribution within the EML is most important. So, the most important parameter that affect the interface degradation is the exciton distribution within the EML or charge balance within the layer. When the exciton populates close to the ETL/EML interface, it is most probable that high energy negative polarons generates, which leads dissociative channels of chemical bonds or other chemical reactions.

Yours sincerely,

Professor Ian S. Gilmore FMedSci

NPL Senior Fellow

Director of National Centre of Excellence in Mass Spectrometry Imaging

Email: ian.gilmore@npl.co.uk

Tel: +44 (0)208 943 6922